# The Role of Oil Prices in Exchange Rate Movements: The CIS Oil Exporters

**Fakhri Hasanov** [1,2,3,*]**, Jeyhun Mikayilov** [4,5]**, Cihan Bulut** [6]**, Elchin Suleymanov** [7,8] **and Fuzuli Aliyev** [7]

[1] King Abdullah Petroleum Studies and Research Center, P.O. Box 88550, Riyadh 11672, Saudi Arabia
[2] Research Program on Forecasting, Economics Department, The George Washington University, 2115 G Street, NW, Washington, DC 20052, USA
[3] Institute of Control Systems, B. Vahabzade Street 9, Baku AZ1141, Azerbaijan
[4] Department of Statistics, Azerbaijan State University of Economics, Istiqlaliyyat Str., 6, Baku AZ1001, Azerbaijan; ceyhun.mikayilov@kapsarc.org
[5] Institute for Scientific Research on Economic Reforms, 88a, Hasan Bey Zardabi Avenue, Baku AZ1011, Azerbaijan
[6] Faculty of Business and International Relations, Vistula University, 3 Stokłosy Str., Warsaw 02-787, Poland; c.bulut@vistula.edu.pl
[7] Department of Finance, Baku Engineering University, Hasan Aliyev 120, Khirdalan AZ0101, Azerbaijan; elsuleymanov@qu.edu.az (E.S.); faliyev@qu.edu.az (F.A.)
[8] The Institute of Economics, Azerbaijan National Academy of Sciences, H. Javid pr., 115., Baku AZ1001, Azerbaijan
[*] Correspondence: fakhri.hasanov@kapsarc.org; Tel.: +966-540-900-964

**Abstract:** Undoubtedly, oil prices play a crucial role in the macroeconomic performances of oil-exporting developing countries. In this regard, the exchange rate is one of the key macroeconomic indicators worthy of investigation. Existing literature shows that world oil prices play an important role in the appreciation of the exchange rates of oil-exporting developing countries. However, only a few studies have examined this issue by considering all three oil-exporting countries of the Commonwealth Independent States, namely Azerbaijan, Kazakhstan and Russia, together. In order to fill this gap and given the increasing importance of these economies in the world's energy markets, this paper examines the role of oil prices in the movement of real effective exchange rates of the above-mentioned CIS countries. We applied the autoregressive distributed lag bounds testing method with a small sample bias correction to the data of these countries over the 2004Q1–2013Q4 period. The estimation results indicate that oil prices are certainly a main driver behind real effective exchange rate appreciation in the selected economies. Moreover, estimations show that productivity, to some extent, can also lead to the appreciation. The policy implication of this research is that an appreciation of the real exchange rate is harmful for the exports of non-oil goods and services in these countries. Since oil prices lead to the appreciation mainly through higher domestic prices, which is a result of tremendous public spending, decision-makers should reconsider the prevailing fiscal policy to make it much more counter-cyclical.

**Keywords:** real effective exchange rate; oil price; CIS oil exporters; autoregressive distributed lag bounds testing approach

**JEL Classification:** F31; F41; C31; P24; Q43

## 1. Introduction

During the last two decades, the economies of CIS countries have been subject to great transformations. A deep collapse in production at the beginning of the transition process was followed by a recovery in the late 1990s. After gaining independence, development in the oil mining industry and oil contracts signed with international oil companies triggered the CIS oil exporters, namely, Russia, Kazakhstan and Azerbaijan, and increased their importance in the world's energy markets (Hasanov et al. 2016). Since these countries are located in the Caspian basin neighborhood and they were the former Soviet Union republics, they have many common socio-economic characteristics. One common feature is the sensitivity of their economies to the changes in oil prices, or more generally, to the various oil shocks. Almost all of the economic activities and "behaviors" of key macroeconomic indicators are somehow related to the oil sector within these countries and in their relationships with other countries (Kose and Baimaganbetov 2015; Kaplan and Aktash 2016; Hasanov 2013 inter alia).

The change in oil prices has been followed by large changes in very short periods of time, with the 2008 oil price fluctuations being a good example. The oil price per barrel was 90.8 USD at the beginning of the year; but, by July, the price reached a historic peak of 132.5 USD, and by December in the same year, the price had fallen to 41.5 USD. The coefficient of the variation of oil prices during the 2004Q1–2013Q4 period (the estimation period of the present study) was 29.1%, indicating that there is substantial volatility in oil price movements. As a consequence, the income of oil exporters can substantially vary within a short period of time. The huge fluctuations in the income of the countries reliant on oil export revenues might well cause considerable effects on the rest of their economies.

Exports, which represent one of the main pillars of the economy, also "receive their share of setbacks" from the oil price fluctuations in the oil-exporting countries. The real exchange rate, one of the drivers of exports, reflects the level of a country's competitiveness. It is one of the key indicators to be analyzed in the oil price-effects framework. It is a common perception that the real exchange rate has major impacts on oil exporters' overall macroeconomic performances. As discussed in a number of studies (e.g., Corden and Neary 1982; Corden 1984; Wijnbergen 1984; Buiter and Purvis 1983; Bruno and Sachs 1982; Enders and Herberg 1983; Edwards 1985), there is a strong relationship between the resource price and the real exchange rate. The impact of the oil price on the real exchange rate in oil-exporting countries might occur through different channels (Hasanov 2010). There is a vast literature investigating these effects in the oil exporters (see: Koranchelian (2005) for Algeria; Zalduendo (2006) for Venezuela; Issa et al. (2006) for Canada; Habib and Kalamova (2007) for Norway, Saudi Arabia and Russia; Oomes and Kalcheva (2007) for Russia; Korhonen and Juurikkala (2009) for nine OPEC countries; Jahan-Parvar and Mohammadi (2008) for fourteen oil exporters; and Hasanov (2010) for Azerbaijan). However, to the best of our knowledge, there is not a time series study that investigates this relationship, by including all of the above-mentioned CIS countries in its analysis.

The above-mentioned, resource-rich countries were able to pursue quite stable exchange rate policies from the early 2000s, right up to 2013, but could not continue in this direction, for the last few years, due to the presence of dramatic changes in oil prices. Subsequently, it would be relevant to explore the relationship between oil prices and real exchange rates in the case of Russia, Kazakhstan and Azerbaijan. Thus, the objective of the present study is to analyze the effect of oil price on the real exchange rate in the above-mentioned countries.

We will, in particular, address the following research questions for the selected economies: (a) Is there a long-run relationship between oil prices and exchange rates? (b) How long does it take to converge from the short-run deviation to the long-run equilibrium?

A brief look at the economies of these three countries will provide some rationale for our study.

Exports correspond to about a third of Russian GDP, and roughly half of these export revenues come from energy. According to the Energy Information Agency (EIA), oil and natural gas sales accounted for 68% of Russia's total export revenues in 2013. A large part of Russia's federal budget was based on oil and natural gas activities. As indicated by the Ministry of Finance, 50% of Russia's federal budget revenue in 2013 came from mineral extraction taxes and export customs duties on

oil and natural gas (EIA 2014). In these circumstances, one can expect to find a strong relationship between oil price and foreign currency inflow for Russia.

Kazakhstan possesses extensive natural resources and relies heavily on revenues from the export of primary commodities, in particular petroleum and natural gas. Kazakhstan is ranked 11th in the world, in terms of proven oil reserves, and is the second largest oil producer among the former Soviet Republics, after Russia, having produced nearly 1.7 million barrels per day in 2014 (KCCG 2016). Kazakhstan's dependence on oil revenues suggests that the economy is vulnerable to oil prices.

Azerbaijan has been increasing its oil extraction and exportation, resulting in a huge inflow of oil revenues into the country. Based on the Asian Development Bank report (Asian Development Bank 2014), for 2010, approximately 80% of all budget revenue derived from the oil industry, through the State Oil Fund of the Republic of Azerbaijan (SOFAZ) and taxes on the oil sector. In 2013, SOFAZ alone accounted for 58% of the state budget revenues. The economy experiences a persistent appreciation of the exchange rate, which hurts the competitiveness of the non-oil tradeable sector (Hasanov 2013 inter alia). This appreciation is closely related to oil prices and, subsequently, oil revenues.

As can be seen from the brief descriptions of the economies of interest, one can surmise that the real effective exchange rate heavily depends on oil prices.

Applying the autoregressive distributed lag bounds testing method, which has certain advantages in the case of small samples, to the data of the countries over the period of 2004Q1–2013Q4, we find that oil prices are amongst the main drivers of appreciation of the real exchange rate in the long-run.

The contribution of this study to the existing literature is two-fold. First, to the best of our knowledge, this is the first time series study that investigates the effect of oil prices on exchange rates by considering all three CIS oil exporters together. One may think that there are some panel studies that consider all three economies along with other oil exporters. However, as pointed out by many studies, panel analysis assumes that the response of dependent variables to explanatory variables is homogenous for all panel members, and therefore, it does not capture country-specific features of a research question at hand (see Hsiao 2003; Hsiao 2007; Hasanov et al. 2016 inter alia). Second, our paper addresses the issue of non-stationarity with and without breaks and cointegration properties of the data using two different methods, especially considering small sample bias correction in empirical analysis.

The results of this study are helpful in understanding exchange rate movements in the selected countries. Therefore, the study has useful policy implications, bearing in mind the recent period of oil price slowdown and exchange rate devaluations.

The structure of the remaining part of the paper is as below. The Literature Review section reviews relevant literatures, while the Theoretical and Modeling Framework describes the functional form specification between real exchange rates and the movements of oil prices. The employed econometric methodology and data used are described in the Data and Econometric Method section. The Empirical Estimations and Discussion section comprises the results of econometric estimations and their interpretations. The main findings of the study, along with proposed policy recommendations, are discussed in the Concluding Remarks section. The Reference section consists of a list of reviewed literature works.

## 2. Literature Review

There are a number of studies that explore the impact of oil prices on exchange rates for countries and/or country groups of oil exporters. Since our research objective is to investigate this relationship in the three CIS oil-exporting countries, in order to save space, we will only review here those studies that are devoted to the selected countries. The reviewed literature is presented in Table 1.

What follows is a brief discussion of the drawbacks of the reviewed studies in Table 1. Jahan-Parvar and Mohammadi (2008) only have used the real oil price as an independent variable, which can be viewed as a potential weakness in this study, since it does not take into account the fact that there are other drivers behind exchange rates, which might cause misspecification problems.

**Table 1.** Studies for the CIS oil-exporting developing economies.

| Study | Period | Country/Region | Methodology | Oil Price Coefficient | Productivity Coefficient |
|---|---|---|---|---|---|
| (Kuralbayeva et al. 2001) | 1994M1–2000M4 | KZ | VECM | 0.23 | 0.36 |
| (De Broeck and Slok 2001) | 1993–1998 | Panel of countries including AZ, KZ and RU | PMG | −0.55 * (for the group where AZ, KZ and RU were included) | 0.57 (for the group where AZ, KZ and RU were included) |
| (Spatafora and Stavrev 2003) | 1995Q1–2002Q3 | RU | ECM | 0.31 | 1.3 |
| (Egert 2005) | 1991M1–2003M11 | Country group (RU included) | DOLS, ARDL. Panel DOLS, MGE | −0.18 to −0.06 (for panel) | −1.1 (for RU) |
| (Sosunov and Zamulin 2006) | 1998–2005 | RU | CGE | 0.25 | Not used |
| (Habib and Kalamova 2007) | 1995Q1–2006Q2 (for RU) 1980Q1–2006Q2 (for other countries) | RU, Norway and Saudi Arabia | VECM | 0.50 | 0.82 |
| (Oomes and Kalcheva 2007) | 1995M1–2005M12 | RU | VECM | 0.50 and 0.58 | 1.08 and 0.90 |
| (Egert and Leonard 2007) | 1996–2005 | KZ | OLS, DOLS, ARDL | Not reported | Not reported |
| (Jahan-Parvar and Mohammadi 2008) | 1970–2007 | 14 oil exporters, including RU | ARDL | 1.12 (for RU) | Not used |
| (Egert 2009) | 1999–2006 | CIS (including AZ, KZ and RU) | Panel | Not reported | Not used |
| (Benedictow et al. 2010) | 1995Q1–2008Q1 | RU | OLS | 0.19 | Not used |
| (Hasanov 2010) | 2000Q4–2007Q4 | AZ | ARDL | 0.75 | 2.00 |
| (Babayev 2010) | 1999–2009 | AZ | VAR | Not reported | Not reported |
| (Ito 2010) | 1994Q1–2009Q3 | RU | VECM | 0.17 | Not used |
| (Basher et al. 2012) | 1988M1–2008M12 | Group of countries, including KZ | SVAR | Not reported | Not reported |
| (Mironov and Petronevich 2015) | 2002M5–2013M4 | RU | VECM | 0.2 | Not reported |
| (Kose and Baimaganbetov 2015) | 2000M1–2013M12 | KZ | SVAR | Not reported | Not reported |
| (Bouoiyour et al. 2015) | 1995Q1–2009Q3 | RU | ARDL, Wavelet and FDA | Not reported | Not reported |
| (Kaplan and Aktash 2016) | 1995–2014 | Five countries (including RU) | CCEMG, CCEP | −0.16 | Not used |
| (Aleksandrova 2016) | 2012–2016 | Caucasus and Central Asia, including AZ and KZ | Descriptive analyses | Not reported | Not reported |
| (Blokhina et al. 2016) | 2000M1–2016M1 | RU | OLS | 0.1 | Not used |

Notes: VECM = Vector Error Correction Model, ARDL = Autoregressive Distributed Lagged Model, VAR = Vector Autoregression, SVAR = Structural VAR, CCEMG = Common Correlated Effects Mean Group, CCEP = Common Correlated Effects Pooled, PMG = Pooled Mean Group, DOLS = Dynamic Ordinary Least Squares, CGE = Computable General Equilibrium, FDA = Frequency Domain Approach, OLS = Ordinary Least Squares, CIS = Commonwealth of Independent States, AZ = Azerbaijan, KZ = Kazakhstan, RU = Russia; * the coefficient was found to be insignificant.

Since the main objective in Oomes and Kalcheva (2007) was examining Dutch disease symptoms, including exchange rate appreciation, they have not put special attention on the exchange rate. Furthermore, the study did not consider the small sample bias issue, and the speed of adjustment coefficient was not reported, though the study makes use of relevant specifications and an econometric approach.

The main research focus of Benedictow et al. (2010) was constructing a macroeconometric model for the Russian economy, rather than exploring the response of the exchange rate to oil price change in much detail. Moreover, the study has not taken into account the small sample bias issue.

Basher et al. (2012) analyzed the responses of some economic indicators to the shocks on their main drivers, including the response of the exchange rate to the oil price for a panel of countries, including Kazakhstan. Likewise, Kose and Baimaganbetov (2015) have investigated the direction, significance and duration of the impulse-response functions for Kazakhstan. The impulse-response

analyses were used in the studies as an analytical tool, and therefore, the magnitude of the effects (coefficients and or elasticities) was not estimated.

Although Babayev (2010) has not estimated the slope coefficient, or elasticity, of the real effective exchange rate with respect to the real oil price and has not addressed the long-run equilibrium and the adjustment process of the relationships; it is one of the very few papers devoted to Azerbaijan.

Ito (2010) and Mironov and Petronevich (2015) neither addressed the small sample bias issue, nor reported the speed of adjustment parameters.

Kaplan and Aktash (2016) concluded that an increase in the oil price causes the appreciation of the domestic currency in the case of Russia. However, the study has not reported any Russian-specific parameters.

Aleksandrova (2016) restricted itself with descriptive analyses and has not employed any econometric estimation methods.

The main drawback of (Blokhina et al. 2016) is that it has not taken into account the non-stationarity properties of the variables, and hence, the results might be spurious.

Finally, as can be noticed from Table 1, most of the studies were devoted to Russia, rather than considering all three CIS countries together. In this regard, Azerbaijan and then Kazakhstan are less studied countries in the time series context.

We shall address all of the above-discussed issues in this study.

## 3. Theoretical and Modeling Framework

### 3.1. Real Exchange Rate Equation

The theoretical framework is a standard model for investigating the exchange rate in small open economies depending on commodity exports developed by Cashin et al. (2004). They have conceptually adopted a two-variable exchange rate equation for commodity-exporting countries, where the exchange rate is a function of commodity price as a measure of the terms of trade. Following the same conceptual framework, Habib and Kalamova (2007) have developed the framework for oil exporting economies, where the Real Exchange Rate (RER) is a function of Oil Price (OILP) and the Productivity differential measured by the relative price ratio (PROD) given below:

$$RER = f(OILP, \ PROD) \tag{1}$$

For the purpose of econometric estimation in the time series context, Equation (1) can be expressed as follows:

$$rer_t = \beta_0 + \beta_1 \times oilp_t + \beta_2 \times prod_t + \varepsilon_t \tag{2}$$

where small letters denote a natural logarithmic expression of given variables and $\varepsilon_t$ is an error term.

It is noteworthy that other studies have also used such a parsimony specification for examining the exchange rate in oil-exporting developing economies; for example, Hasanov (2010) for Azerbaijan, Korhonen and Juurikkala (2009) for 14 oil exporters including Russia; Egert (2009) for the former Soviet Union countries, including Russia, Kazakhstan and Azerbaijan; Kaplan and Aktash (2016) for five countries, including Russia; even Benedictow et al. (2010), Ito (2010) for Russia, Jahan-Parvar and Mohammadi (2008) for 14 oil exporters, including Russia, Nikbakht (2010) for seven OPEC members, Ahmad and Hernandez (2013) for 12 oil producers and consumers have used only oil price to explain the exchange rate in their empirical analysis.

Equation (2) is especially straightforward to employ when the number of observations is small, as it has only two explanatory variables to describe the behavior of an exchange rate. Additionally, Rautava (2004) among others underline that in the case of a small number of observations, there is the need to keep specifications as parsimonious as possible, in order to allow for the parameters to be estimated properly.

### 3.2. Movements of Oil Prices and REERs: Choosing the Period of Analysis

Figure 1 below illustrates the historical time path of oil prices and the Real Effective Exchange Rate (REER) of the selected countries. As shown in the figure, the REERs of these countries appreciated at the beginning of independence (approximately in the years 1991–1996) after the Soviet Union collapsed, because of the so-called transition process, and since then, have experienced a depreciation. The depreciation period was quite long (till 2003 and 2004) in Kazakhstan and Azerbaijan, compared to that in Russia. The depreciation period has been followed by a second appreciation period, where the appreciation was quite persistent over a long time. As marked with the dashed vertical lines in the figure, the period started from 1998Q4 for Russia, while in Kazakhstan and Azerbaijan, it started from 2003Q2 and 2004Q1, respectively.

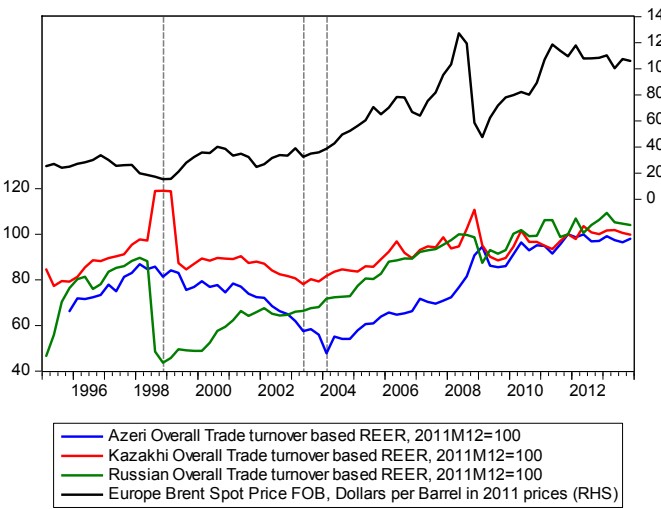

**Figure 1.** Time profile of oil prices and Real Effective Exchange Rates (REERs).

We have tried to match the movements of the oil prices and exchange rates of these countries and discovered that there was no consistent movement between the variables before the second period of appreciation.

The oil prices continuously increased till 1997Q1 and then declined on a regular basis up to 1999Q1, and this pattern (up and down) continued till 2002Q1. Kazakhstani and Azerbaijani exchange rates increased till the end of 1998 and then declined up to 2003Q2 and 2004Q1, respectively. For the Russian exchange rate, the movement was different from the other two countries' cases: an increase up to the middle of 1998, then a sharp drop because of the financial crisis and recovery since 1999Q1.

However, one can easily observe in Figure 1 that the exchange rates quite closely follow the world's oil price development in the second period of appreciation. This picture in itself would suggest that world oil prices can be considered as one of the driving factors of appreciation of the selected counties' REERs.

In order to make the country-specific estimation results comparable to each other, we are analyzing the period of 2004Q1–2013Q4 in this study. Otherwise, the estimation results cannot be compared, if we run our estimations for the different periods for the specific countries, i.e., 1999Q1–2013Q4 for Russia, 2003Q2–2013Q4 for Kazakhstan and 2004Q1–2013Q4 for Azerbaijan. Although we start our estimations from 2004Q1, there are 40 observation points, and additionally, we employ small sample bias corrections in the estimations.

## 4. Data and Econometric Method

This section introduces data used in our empirical analysis first and then describes the method of empirical analysis.

*4.1. Data*

Research covers quarterly data over the period of 2004Q1–2013Q4 and includes: Real Effective Exchange Rates (REERs), real Oil Price (OILP), and Productivity differential (PROD). The following is a detailed description of the variables.

Real Effective Exchange Rate (REER) is a multilateral consumer price index based on the real effective exchange rate of a domestic currency, relative to its main trading partners, which is calculated as below:

$$REER = NEER \times \frac{CPI^D}{CPI^F}$$

whereas $NEER$ and $CPI^D$ are the Nominal Effective Exchange Rate and Consumer Price Index of a domestic economy; $CPI^F$ is the weighted average Consumer Price Index of main trading partners.

It is defined in terms of foreign currency per unit of domestic currency, so that an increase in REER means an appreciation of the domestic currency. Note that REERs, for the selected countries, are calculated by the below-mentioned sources. REERs for Azerbaijan and Kazakhstan are collected from the statistical bulletins of the Central Bank of Azerbaijan and National Bank of Kazakhstan. The Russian REER is taken from the Bank for International Settlements.

Real Oil Price (OILP): This variable is calculated as the Europe Brent Spot Price FOB (Free on Board) for Crude Oil, Dollars per Barrel in Nominal terms (NOILP), deflated by the Consumer Price Index of the USA (CPI_US). NOILP and CPI_US are retrieved from the U.S. Energy Information Administration and the Organization for Economic Co-operation and Development.

Productivity differential (PROD): Following Alberola et al. (1999) and Hasanov (2010), we have used the relative price of non-traded goods to traded goods as a measure of productivity differential. These studies discuss that usually, the price of non-traded goods is represented by the Consumer Price Index, while the Producer Price Index measures the price of traded goods. We have calculated the productivity differential as follows:

$$PROD = \frac{CPI^D/PPI^D}{CPI^W/PPI^W} \times 100$$

$CPI^D$ and $PPI^D$ are the Consumer Price Index and Producer Price Index for a domestic economy. $CPI^W$ and $PPI^W$ denote the same variables for the rest of the world. CPI and PPI for Azerbaijan, Kazakhstan and Russia have been taken from the State Statistical Committee and Central Bank of Azerbaijan, the Agency of Statistics of the Republic of Kazakhstan and Federal State Statistics Service of the Russian Federation, respectively. Those for the rest of the world have been collected from the International Monetary Fund and Haver Analytics.

In order to keep consistency, all variables have been re-based to 2011Q4. Note that in the empirical analysis, we use the natural logarithmic expressions of the variables, which are denoted with small letters: *reer*, *oilp*, *prod*. Furthermore, note that *_az*, *_kz*, and *_ru* attached to the end of a variable name indicate that a given variable refers to Azerbaijan, Kazakhstan and Russia, respectively.

Just for illustrative purposes, the figures below portray the time profile of the logarithmic level and growth rate of countries' variables and oil prices over the period of 2004Q1–2013Q4.

*4.2. Econometric Method*

In this sub-section, first, we present the Augmented Dickey–Fuller (ADF) and Zivot and Andrews (ZA) tests for unit root and then discuss the Autoregressive Distributed Lag Bounds Testing (ARDL) approach to cointegration. Finally, we describe small sample bias correction, which is not addressed in the prior studies for the selected economies.

4.2.1. Unit Root Test

Before conducting a cointegration analysis by applying a cointegration method, the order of integration of the variables has to be examined by means of the Unit Root (UR) test. We employ the

ADF ([Dickey and Fuller 1981](#)) test for this purpose. The test maintains the null hypothesis of the non-stationarity of a given time series.

For a variable *y*, ADF statistics apply the *t*-ratio on $b_1$ from the regression:

$$\Delta y_t = b_0 + \psi trend + b_1 y_{t-1} + \sum_{i=1}^{k} a_i \Delta y_{t-i} + \varepsilon_t \qquad (3)$$

Here, $\Delta$ and $k$ stand for the first difference operator and number of the lags, respectively, $b_0$ is a constant term, *trend* and $\varepsilon_t$ are the linear time trend and white noise residuals, $\psi$ is coefficient of trend, $b_1$ is the coefficient of lagged level of the dependent variable, while $\alpha_i$s are coefficients of the lagged differenced terms, *i* is the lag order.

Due to space limitation, we are not going to discuss this test here. [Dickey and Fuller (1981)](#), [Stock and Watson (1993)](#), [Dolado et al. (1990)](#), [de Brouwer and Ericsson (1998)](#) and [Enders (2010)](#), pp. 237–39), among others, discuss the advantages and disadvantages of the univariate UR tests, in particular ADF. As can be seen from Figures 2 and 3, some of the variables might have a structural break. Therefore, in order to address this issue, we also have employed the [Zivot and Andrews (1992)](#) unit root test. Due to the space limitation, we will not describe the test here. The details and helpful overview of it can be found in [Zivot and Andrews (1992)](#) and [Perron (2006)](#) inter alia.

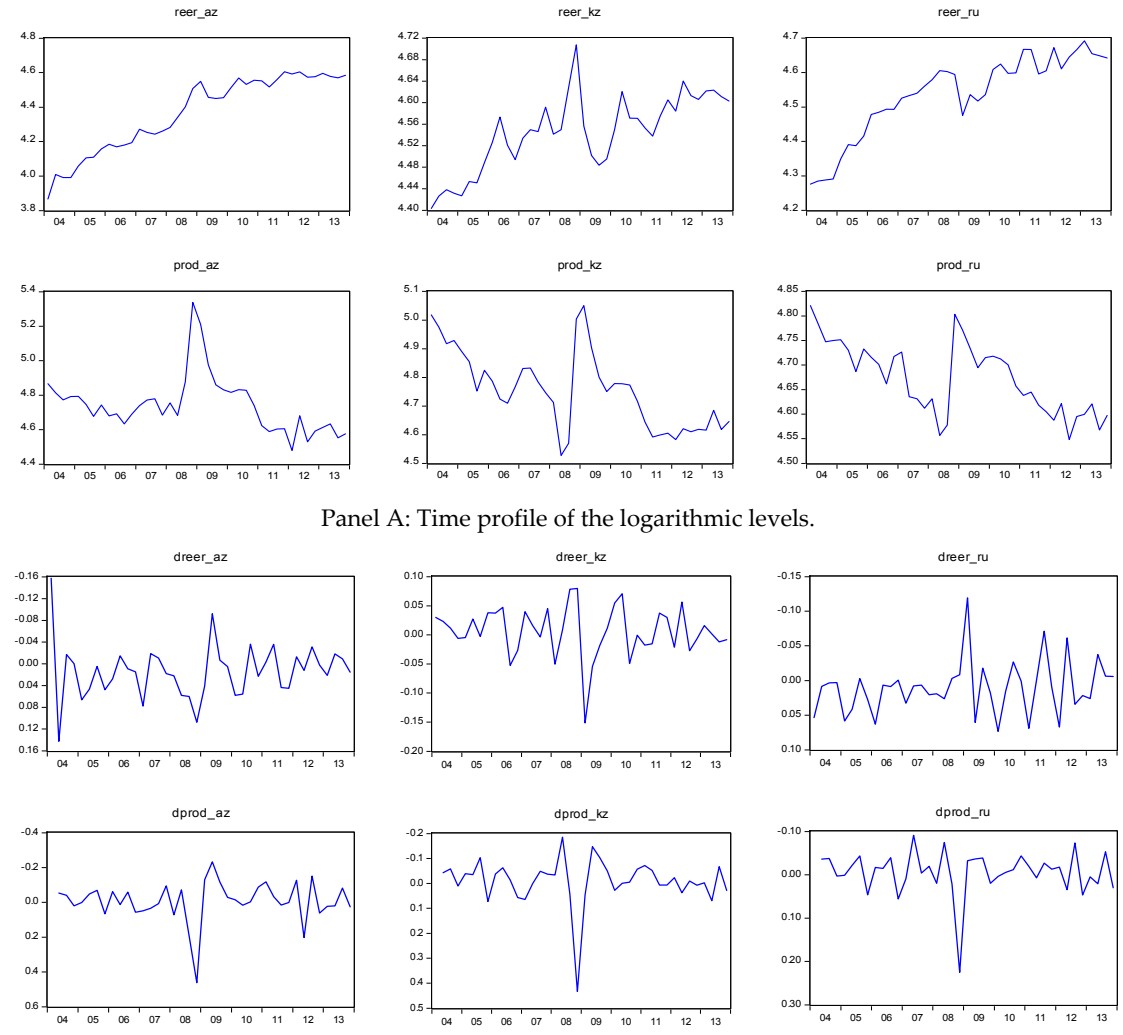

Panel A: Time profile of the logarithmic levels.

Panel B: Time profile of the growth rates.

**Figure 2.** Time profiles of log levels and growth rates of the variables.

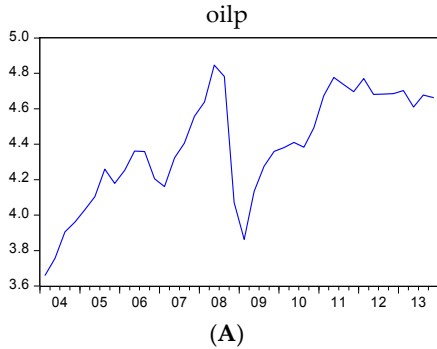
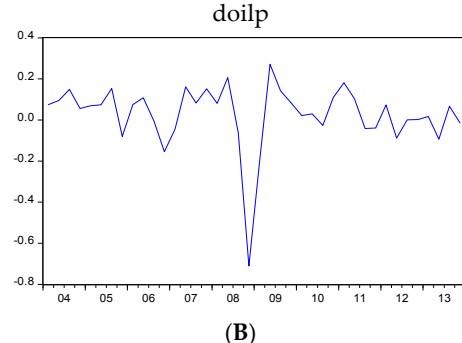

(**A**)  (**B**)

**Figure 3.** Time profile of log levels and growth rates of the real oil prices. (**A**) Time profile of the logarithmic levels. (**B**) Time profile of the growth rates.

### 4.2.2. Autoregressive Distributed Lag Bounds Testing Approach

A cointegration approach that we are going to apply to the data is the ARDL approach developed by Pesaran et al. (2001), and Pesaran and Shin (1999). This approach outperforms its counterparts, in the case of small samples among its other advantages, such as: being easy to perform just by using OLS; estimating long- and short-run coefficients, simultaneously; applicability regardless of regressors being I(1) and I(0) or a mixture of them (Pesaran et al. 2001; Oteng-Abayie and Frimpong 2006; Sulaiman and Muhammad 2010). When one considers that we have a relatively small number of observations, the approach is more appropriate for our empirical analysis.

As Pesaran et al. (2001) describe, the approach has the following stages:

(a)  Construction of an unrestricted Error Correction Model (ECM).

$$y_t = c_0 + \theta y_{t-1} + \theta_{yxx} x_{t-1} + \sum_{i=1}^{n} \omega_i \Delta y_{t-i} + \sum_{i=0}^{n} \varphi_i \Delta x_{t-i} + u_t \tag{4}$$

where $y$ is a dependent variable, while $x$ is an explanatory variable; $u$ denotes white noise errors; $c_0$ is for a drift coefficient; $\theta_i$ indicate long-run coefficients, while $\omega_i$ and $\varphi_i$ are short-run coefficients.

Note that one of the main issues in the ARDL estimations is to correctly specify the lag length of the first differenced right-hand side variables, as finding a cointegrating relationship between variables is sensitive to this (Pesaran et al. 2001, p. 23). Following Pesaran et al. (2001), among others, the optimal lag length can be specified by minimizing the Akaike and Schwarz information criteria, whilst removing the serial autocorrelation of residuals. In small sample cases, it is advisable to rely on the Schwarz information criterion (Pesaran and Shin 1999; Fatai et al. 2003).

(b)  Once an unrestricted ECM is constructed, the existence of a cointegrating relationship can be tested. The Wald-test (or the *F*-test) on the $\theta_i$ coefficients above is performed for this purpose.

The null hypothesis of no cointegration is stated as: H$_0$: $\theta_1 = \theta_2 = \theta_3 = 0$; while an alternative hypothesis of cointegration is: H$_1$: $\theta_1 \neq \theta_2 \neq \theta_3 \neq 0$.

If the computed/sample *F*-statistic is greater than the upper bound of the critical value for a given significance level, then the null hypothesis of no cointegration can be rejected. In the same vein, the null of no cointegration cannot be rejected, if the sample *F*-statistic is smaller than the lower bound of the critical value for a given significance level. As a third case, the sample value may fall between critical values of upper and low bands, and in such a case, the test results are inconclusive.

It is important to note that the *F*-statistics in the ARDL cointegration test have a non-standard distribution. Therefore, the conventional critical values of *F*-distribution are not valid anymore, and critical values of the *F*-distribution have to be taken from the table, which is developed by Pesaran and Pesaran (see Pesaran and Pesaran 1997, or Pesaran et al. 2001).

If $\theta$ is statistically significant and negative, then it can be concluded that the cointegrating relationship is stable. In other words, short-run deviations from the long-run equilibrium path are temporary and converge towards it.

(c)   The long-run coefficients can be estimated/calculated, if the cointegrating relationship found among the variables as a result of the previous stage. Note that these coefficients can be calculated based on Equation (4) by either applying a Bewley transformation (Bewley 1979) or manually setting $c_0 + \theta y_{t-1} + \theta_{yxx} x_{t-1}$ to zero and solving it for $y$ as follows:

$$y = -\frac{c_0}{\theta} - \frac{\theta_{yxx}}{\theta} x + u \tag{5}$$

### 4.2.3. Small Sample Bias Correction in the ARDL Approach

Note that Pesaran and Pesaran  (1997) have calculated the upper and lower critical values of the F-distribution by using large sample sizes of 500 and 1000, as well as 20,000 and 40,000 replications, respectively. However, Narayan (2005) argues that these critical values are not accurate for small sample sizes, as they are calculated based on large sample points (Narayan 2004; Narayan 2005). As a matter of fact, he has compared the critical value generated, based on 31 observations, with those values reported in Pesaran et al. (2001), in the case of four regressors and at a 5% significance level. He has revealed that the critical value (3.49) from Pesaran and Pesaran  (1997) is 18.3% lower than the critical value (4.13) that he has calculated. Hence, he has calculated critical values for small sample sizes ranging from 30 through to 80 data points (see Narayan 2005). As a small sample correction, we are going to use those critical values in our ARDL cointegration test.

Note briefly that we also employ the Dynamic Ordinary Least Squares (DOLS) method for a robustness check, as discussed in Section 5.1.

## 5. Empirical Estimations and Discussion

Consistent with the methodological section, we first examined the integration properties of the variables, by means of the ADF test. Note that we used Equation (3) where the intercept and trend are included for testing purposes. Our justification for having such a trend in our testing procedures is as follows: as explained in econometric theory, if the trend is a part of the data generating process and we drop it from our test equation, then we will have biased results, which is a serious problem. By way of contrast, if the trend is not a part of the data-generating process and we have it in the equation redundantly, then we only lose just one degree of freedom. The ADF and ZA test results are given in Table 2 below.

According to the ADF test statistic, the log level of the oil price is trend-stationary at the 5% significance level. However, the $b_1$ coefficient in Equation (3) for the oil price is $-0.40$, and thus, the autoregressive coefficient is 0.60, which is closer to unity than zero, which may be indicative of the unit root process[1]. Moreover, a graphical inspection of the log level and growth rate of the oil price in Figure 3 suggests that the series is an I(1) process rather than trend-stationary. Additionally, the first difference of the series is highly stationary, and the auto-regressive coefficient is equal to 0.06. Both highly suggest that the first difference of roilp is stationary. In addition, the results of the ZA test also show that roilp is an I(1) process.

According to the ADF and ZA tests results, Azerbaijani and Russian real effective exchange rates follow an I(1) process. In other words, they are non-stationary at their log level, but stationary at their growth rates. The same conclusion is also true for the Kazakhstani real effective exchange rate,

---

[1]   Equation (3) is a transformed version of the unit root test equation (random walk with drift), where $p$ is equal to $b_1 + 1$. $p$ is an auto-regressive coefficient. If it is unity (or close to unity), it means that a given series has a unit root. By way of contrast, if it is zero (or close to zero), it means that the process does not have a unit root process.

although the ADF test results suggest that it is a trend-stationary process at the 10% significance level. However, the ZA test results, as well as further analysis (graphical illustration, magnitude of $b_1$ coefficient being $-0.44$) decisively show that the series is an I(1) process.

**Table 2.** The unit root test results.

| Variable | At the Level | | | | At the First Difference | | | |
| | ADF | | ZA | | ADF | | ZA | |
| | *k* | **Actual Value** | *k* | **Actual Value** | *k* | **Actual Value** | *k* | **Actual Value** |
|---|---|---|---|---|---|---|---|---|
| *roilp* | 1 | $-3.91$ ** | 1 | $-4.60$ | 1 | $-5.89$ *** | 1 | $-8.01$ *** |
| Panel A: Azerbaijan | | | | | | | | |
| *reer_az* | 0 | $-2.08$ | 0 | $-3.67$ | 0 | $-6.77$ *** | 0 | $-7.33$ *** |
| *prod_az* | 0 | $-2.34$ | 0 | $-3.89$ | 0 | $-5.88$ *** | 2 | $-8.31$ *** |
| Panel B: Kazakhstan | | | | | | | | |
| *reer_kz* | 0 | $-3.37$ * | 3 | $-2.88$ | 1 | $-5.97$ *** | 4 | $-8.53$ *** |
| *prod_kz* | 1 | $-4.40$ *** | 1 | $-4.81$ | 1 | $-6.25$ *** | 2 | $-7.90$ *** |
| Panel C: Russia | | | | | | | | |
| *reer_ru* | 0 | $-2.62$ | 0 | $-1.81$ | 1 | $-6.45$ *** | 1 | $-6.55$ *** |
| *prod_ru* | 0 | $-3.44$ * | 0 | $-1.66$ | 0 | $-6.89$ *** | 2 | $-7.24$ *** |

Notes: Maximum lag order is set to four, and optimal lag order (*k*) is selected based on Schwarz's criterion in the Augmented Dickey–Fuller (ADF) and Zivot and Andrews (ZA) tests; *, ** and *** indicate rejection of the null hypothesis of unit root at the 10%, 5% and 1% significance levels, respectively; the critical values are taken from MacKinnon (1996) for the ADF test and from Vogelsang (unpublished computer program) for the ZA test. Estimation period: 2004Q1–2013Q4.

Both the ADF and ZA tests conclude that the productivity differential variables for Azerbaijan and Russia are non-stationary at the log level and stationary at the growth rate. In the Kazakhstan case, the ADF test suggests trend stationarity, while the ZA test prefers an I(1) process with a structural break. The graphical illustration in Figure 2 is also in favor of an I(1) process. This is usual practice in empirical analysis, that the ADF test can be more biased when the given time series has a structural break.

Thus, as a summary of the unit root exercise, we conclude that all variables are non-stationary at their log level and stationary at their growth rate. In other words, they follow an I(1) process.

We estimated Equation (4) for each country, by setting *n* to be equal to four as a maximum lag length, because in the case of a small number of observations, it is not suggestive to start with a higher lag order. In optimal lag selection, we rely on the Schwarz information criterion (again due to the small number of observations) and the non-existence of serial correlation in the residuals of the selected specification. Note that the estimations have been realized using the EViews 9.5 software package, which automatically chooses the optimal ARDL specification. The found optimal ARDL specifications are (3,0,0), (1,0,1) and (4,4,3) for Russia, Kazakhstan and Azerbaijan, respectively.

We use the above-selected specifications for testing the existence of cointegration among the lagged level variables as the next procedure of the ARDL approach. It is noteworthy that, in order to avoid potential biases caused by the small sample size of the estimations, we use Narayan's (Narayan 2005) critical values, along with Pesaran et al. (2001). Table 3 reports the cointegration test results.

In the Azerbaijani and Kazakhstani cases, there is a cointegrating relationship among the real effective exchange rate, real oil price and productivity differential, at the 1% significance level, according to Pesaran et al. (2001) and Narayan (2005) as a small sample bias correction, respectively. Both tests statistics also reject the null hypothesis of no cointegration among the same variables in Russia at the 5% significance level. Thus, we can reject the null hypothesis of no cointegration, in favor of the alternative hypothesis of cointegration among the variables for all three countries, as a conclusion of the ARDL bounds test, even after a small sample bias correction.

**Table 3.** Cointegration test statistics.

| $F_{Sample}$ | Panel A: Azerbaijan | | Panel B: Kazakhstan | | Panel C: Russia | |
|---|---|---|---|---|---|---|
| | $F^W_{(4,30)}$ = 6.412 | | $F^W_{(4,30)}$ = 13.804 | | $F^W_{(4,29)}$ = 4.907 | |
| Lower and Upper Bound Critical Values in the Case of Two Lagged Level Regressors, Restricted Intercept and No Trend: | | | | | | |
| | Narayan (2005) * | | | Pesaran et al. (2001) ** | | |
| At the 1% significance level: | 4.770 | 5.855 | | 4.99 | 5.85 | |
| At the 5% significance level: | 3.435 | 4.260 | | 3.88 | 4.61 | |
| At the 10% significance level: | 2.835 | 3.585 | | 3.38 | 4.02 | |

Note: * shows lower and upper bound critical values for Bounds test proposed by Narayan (2005); Narayan's (Narayan 2005) critical values above correspond to the 40 observations case. ** shows lower and upper bound critical values for Bounds test proposed by Pesaran et al. (2001). *F* is the *F*-value of the null hypothesis that long-run coefficients in the equation (4) are jointly equal to zero.

The results of the ARDL estimation are given in the Panel A of Table 4.

**Table 4.** The results of ARDL and DOLS estimations.

| | Azerbaijan | | Kazakhstan | | Russia | |
|---|---|---|---|---|---|---|
| | **Panel A: Estimated Long-Run Coefficients** | | | | | |
| Variables | ARDL | DOLS | ARDL | DOLS | ARDL | DOLS |
| *roilp* | 0.264 ** (0.099) | 0.262 *** (0.044) | 0.276 *** (0.083) | 0.285 *** (0.072) | 0.560 *** (0.103) | 0.570 *** (0.119) |
| *prod* | 0.581 *** (0.121) | 0.512 *** (0.052) | 0.237 (0.200) | 0.390 ** (0.149) | 1.150 ** (0.452) | 0.962 * (0.532) |
| | **Estimated Long-Run Coefficients with *nfa* variable** | | | | | |
| *roilp* | 0.458 *** (0.122) | 0.417 *** (0.080) | 0.266 ** (0.098) | 0.657 *** (0.142) | 0.240 *** (0.063) | 0.193 *** (0.067) |
| *prod* | 0.621 *** (0.114) | 0.577 *** (0.056) | 0.223 (0.206) | 1.073 *** (0.242) | 0.399 * (0.207) | 0.277 (0.247) |
| *nfa* | −0.124 * (0.069) | −0.149 (0.091) | 0.004 (0.022) | −0.128 (1.261) | 0.103 *** (0.022) | 0.121 *** (0.018) |
| ECT | −0.50 *** | | −0.63 *** | | −0.33 *** | |
| | **Panel B: Residuals Diagnostics Tests Results** | | | | | |
| | $\chi^2_{SC}(4) = 1.982$ [0.134] | | $\chi^2_{SC}(4) = 0.925$ [0.462] | | $\chi^2_{SC}(4) = 1.008$ [0.419] | |
| | $\chi^2_{HETR}(4) = 0.848$ [0.617] | | $\chi^2_{HETR}(1) = 0.103$ [0.981] | | $\chi^2_{HETR}(3) = 0.6750$ [0.645] | |
| | $JB_N = 0.656$ [0.721] | | $JB_N = 0.426$ [0.808] | | $JB_N = 1.146$ [0.564] | |
| | $F_{FF} = 1.011$ [0.322] | | $F_{FF} = 0.723$ [0.475] | | $F_{FF} = 0.955$ [0.346] | |

Notes: The dependent variable is *reer*; residuals diagnostics tests results are from the selected ARDL specifications; $\chi^2_{SC}$, $\chi^2_{ARCH}$ and $\chi^2_{HETR}$ denote chi-squared statistics to test the null hypotheses of no serial correlation and no heteroscedasticity in the residuals; $JB_N$ and $F_{FF}$ indicate Jarque–Bera and no functional form misspecification statistics to test the null hypotheses of the normal distribution and no functional misspecification, respectively; probabilities are in brackets, and standard errors are in parentheses; ECT is the Error Correction Term, i.e., the residuals of the long-run equation. *, **, *** indicate significance at 10%, 5% and 1%, respectively. Estimation period: 2004Q1–2013Q4.

As can be seen from Panel B of Table 4, the final specifications for all three countries successfully pass the residual diagnostics tests of the autocorrelation, serial correlation, normality and heteroscedasticity, as well as Ramsey RESET misspecification test. The recursive residuals stability test does not find any significant instability, and the specifications are completely stable towards the end of the period, which is desirable[2]. In all specifications above, the Speed of Adjustment coefficients, i.e., the coefficients on ECTs are statistically significant at the 1% significance level, which is indicative of the stable long-run relationship between the real effective exchange rate and the oil price and the productivity differential in all three countries.

Moreover, in all three above-given specifications, the real oil price has a positive impact on the real effective exchange rate at the 1% significance level in Kazakhstan and Russia and the 5% significance level in Azerbaijan. This indicates that the real oil price is one of the main drivers of the real exchange rate appreciation, as expected, theoretically and empirically.

---

2　　The test results can be obtained from the authors upon request.

We can conclude that the productivity differential is also one of the determinants of the real effective exchange rate movement in the long-run, since it is statistically significant at the 1% and 5% levels in the Azerbaijani and Russian cases. The variable can be considered as a one of the main drivers of real effective exchange rate appreciation in Kazakhstan since it is statistically significant at the 5% significance level in the DOLS estimation, though it is not significant in the ADRL specification.

According to the long-run elasticities from the ARDL estimation results, a 1% rise in the real oil price leads to a 0.26%, 0.28% and 0.56% appreciation of Azeri, Kazakh and Russian real effective exchange rates, respectively. The coefficients are quite close to each other in the Azerbaijani and Kazakhstani cases, and higher in the Russian case, which would indicate that the oil price plays a very similar role in the former two economies. In fact, all of these countries follow the same exchange rate policy, which is pegged to the U.S. dollar, along with similar fiscal and monetary policies. Our obtained elasticity of 0.560 for Russia is close to those from Habib and Kalamova (2007) and Oomes and Kalcheva (2007), being 0.50 and 0.50–0.58, respectively. In the Kazakhstani case, our finding is close to the numerical value estimated by Kuralbayeva et al. (2001), being 0.23. Finally, for Azerbaijan, our obtained exchange rate elasticity, with respect to the real oil price, is quite smaller than what was estimated by Hasanov (2010), the only prior time series study reporting the coefficient of interest. One possible explanation for this difference would be the different time periods used. Another possible explanation is that the real effective exchange rate depreciated from 2000–2004Q1, and then appreciated since then, as illustrated in Figure 1. This might cause a structural break, but Hasanov (2010) has not addressed this issue.

The productivity differential can be considered as another source of the real exchange rate appreciation in the selected economies. More precisely, a 1% increase in the productivity differential causes 0.58%, 0.24% and 1.15% appreciation of Azeri, Kazakh and Russian real effective exchange rates, respectively. For the Kazakhstani case, our estimated coefficient slightly differs from that of Kuralbayeva et al. (2001) and De Broeck and Slok (2001), being 0.36 and 0.57, respectively. The difference might be caused by the use of a different variable as a proxy for productivity and time spans. In the Russian case, our coefficient is close to the findings of Spatafora and Stavrev (2003), Habib and Kalamova (2007) and Oomes and Kalcheva (2007), being 1.3, 0.82 and 1.08, respectively. As for Azerbaijan, our estimated elasticity is in line with the finding of De Broeck and Slok (2001), being 0.57 for the panel including Azerbaijan, and significantly differs from that of Hasanov (2010). The difference between Hasanov (2010) and our result might be caused by the reason mentioned above.

According to the Speed of Adjustment coefficients, 50%, 63% and 33% of short-run disequilibrium in the proposed relationship can be corrected towards a long-run equilibrium path during a quarter in Azerbaijan, Kazakhstan and Russia, respectively.

Since the impact of the oil price on the REER of these countries is our main focal interest, we should further investigate it. According to the method of calculation, for REER (see the Data section above), there are two domestic channels that the oil price can cause appreciation through: domestic prices and the nominal effective exchange rate. The latter channel is quite limited in transforming the oil price effects into real exchange appreciation, as all of these countries follow a de facto fixed exchange rate policy to keep a nominal exchange rate at the targeted level. In terms of contrast, the former channel is quite large for such a transformation. In particular, considering that although the de-jure aim of the monetary agencies (central or national banks) in these countries is price stability, they mainly focus on nominal exchange rate targeting and, therefore, do not have an independent monetary policy, according to the "impossible trinity" concept (Agazade et al. 2017 inter alia). Thus, on the one side, there is a poorly independent monetary policy, which is quite limited in being able to curb high domestic prices, and on the other side, there is a dominant fiscal policy, which triggers higher prices in these economies. As Sturm et al. (2009) discuss, fiscal policy is in a dominant position, and monetary policy just deals with the consequences of it in oil-exporting economies. The fiscal policies of these countries are pro-cyclical, and therefore, when the oil price and, thus, oil revenues are high, then there is a fiscal expansion, which results in higher domestic price levels (Sturm et al. 2009,

among others). Hasanov (2013) for Azerbaijan and Behar and Fouejieu (2016) for the panel of oil exporters, including Russia, Kazakhstan and Azerbaijan, show again the dominant position of fiscal policy and the importance of government spending in the macroeconomic functioning. The Asian Development Bank report discusses tremendous government spending in Azerbaijan especially when oil price was higher (Asian Development Bank 2014). Additionally, Hasanov (2013) outlines that such drastic spending results in high price level especially in the non-tradable sector, which is the main source of real exchange rate appreciation. The above-mentioned relations also hold for Russia and Kazakhstan as discussed by the Center for Social and Economic Research (2008) and Egert (2009), among others. Additionally, Hasanov (2011) investigates price level in a booming economy, i.e., Azerbaijan, and concludes that the main reasons of high price level, especially in the non-tradable sector, are oil revenues and oil prices. He further concludes that monetary policy is quite limited for curbing high price level, while fiscal policy has to take measures in order to boost economic growth in the non-oil tradable sector in a given circumstance.

*5.1. Robustness Check*

As a robustness check, we have followed two different ways: method based and specification based.

For the method-based robustness check, we employed the DOLS cointegration method to our data. The use of DOLS can be justified as follows. According to the asymptotic properties, the method is more powerful than the other residual-based methods, such as the fully-modified ordinary least squares and canonical cointegration regression methods (Utkulu 1997 inter alia). DOLS was proposed by Saikkonen (1992) and Stock and Watson (1993) to construct an asymptotically-efficient estimator that eliminates the feedback in the cointegrating system using lags and leads of regressors assuming that adding lags and leads of the differenced regressors soaks up all of the long-run correlation between the residuals of the cointegration relationship and innovations among the regressors. The details of the model can be found in Saikkonen (1992) and Stock and Watson (1993).

The results of the DOLS estimations are given in Table 4. As can be seen from Panel A in the table, with some exceptions for Kazakhstan, the DOLS results are very close to that of ARDL results in all three country cases. This can be considered as a robustness of the estimation results, which does not change regardless of the estimation methods employed.

For the robustness check of the functional specification used, according to the anonymous referees' suggestion, we have included Net Foreign Assets (NFA) in our main specification and tested whether it can provide an additional explanatory power in explaining the behaviors of the real effective exchange rate in the countries under consideration, i.e., Russia, Kazakhstan and Azerbaijan. As reported in the bottom part of Panel A in Table 4, estimation results show that for the latter two countries, the variable is insignificant with a negative sign regardless of the estimation methods applied. For the Russian case, inclusion of NFA significantly distorts the coefficients of real oil price and the productivity differential. The magnitudes of the coefficients are significantly lower than those found empirically in previous studies. Habib and Kalamova (2007) point out that the terms of trade improve, net foreign assets increase and government expenditure expands when the oil price rises and vice versa. Moreover, Mironov and Petronevich (2015) show that productivity and net foreign assets are perfectly collinear for Russia. The same thing might be the case between net foreign assets and oil price for Russia and other countries considered here, as well.

Therefore, as a research decision, we excluded NFA from our analysis and consider the exchange rate as a function of oil price and relative productivity. Note that other studies have also used such a parsimony specification for examining exchange rate in oil-exporting developing economies as listed in Section 3.1. Additionally, such a simple specification would be preferable when the number of observations is small, which is the case in our study.

## 6. Concluding Remarks

There are a number of studies investigating the impact of the oil price on the movement of the real exchange rate in oil-exporting developing economies. Although a few time series studies have examined this topic for the CIS oil exporters separately, we are not aware of a study doing so considering all of these countries together. In order to fill this void, we investigated the role of the oil price in appreciation of the real effective exchange rate of Azerbaijan, Kazakhstan and Russia, which have an increasing importance in the world's energy market. We applied the autoregressive distributed lag bounds testing method with a small sample bias correction to the data of these countries over the period of 2004Q1–2013Q4. Estimation results indicate that the oil price is one of the drivers of appreciation of the real exchange rate of the selected economies in the long term. Moreover, it has been found that productivity, to some extent, also leads to the eventual appreciation.

Regarding our research questions outlined in the Introduction section, we found a long-run relationship between the oil price and the exchange rates of the selected countries. In addition to this, we discovered that adjustment from the short-run disequilibrium to the equilibrium path is statistically significant.

The policy implications of this research are that an appreciation of the real exchange rate undermines the competitiveness of exports of non-oil goods and services in these countries. A nominal exchange rate has limited capacity in transforming oil price effects into real exchange rate appreciation because of the domestic currency peg to dollar policy. Therefore, the oil price leads to an appreciation mainly through higher domestic prices. Higher domestic prices are mainly the result of tremendous public spending. Therefore, decision-makers in these economies should reconsider underlying fiscal policy to make it much more counter-cyclical by following some conservative fiscal rules and, thereby, loosening dependence on oil prices and revenues. Although they are not directly derived from our research here, the fiscal policies in the selected countries should take such measures, which target fostering economic growth in the non-oil tradable sector. One of these measures is to reduce the pace of budget spending, as they are basically oriented to the non-oil tradable sector. Another set of measures would aim at using oil revenues to expand economic activity in the non-oil tradable sector, non-oil manufacturing and agriculture, by establishing tax exemption, subsidies, soft line credits, enhancing legislation, eliminating institutional constraints among others. As Hasanov (2011) states, as a policy mix, monetary policy with the aim of price stability and fiscal policy with the purpose of full employment and efficient allocation of economic resources have to be implemented jointly with efficient coordination to eliminate the negative effects of oil prices and, thus, to curb high prices, which lead to the appreciation of the real exchange rate.

**Acknowledgments:** We thank the participants of the 19th conference of the Eurasian Business and Economics Society. We are especially grateful to Hashem Pesaran, a keynote speaker of the conference, for his constructive suggestions as well as to Kazem Falahati and Abd Halim Mohd Noor for their recommendations. Finally, the views expressed in this paper are those of the authors and do not necessarily represent the views of their affiliated institutions.

**Author Contributions:** All authors contributed equally to all aspects of the research reported in this paper.

**Conflicts of Interest:** The authors declare no conflict of interest.

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
