# Peer review of "The Role of Oil Prices in Exchange Rate Movements: The CIS Oil Exporters"

_economies, doi:10.3390/economies5020013_

Round 1

Reviewer 1 Report

Referee Report, economies-183788

The Role of Oil Price in Exchange Rate Movements: the CIS Oil-Exporters Falling Oil Prices: Economic and Financial Implications

This paper uses the Autoregressive Distributed Lag (ARDL) method of Pesaran et al. (2001) to discuss the impact of the oil price on the real effective exchange rates of three CIS countries. As an empirical exercise, it is competently done, but there are a few remaining questions.

1) Why is interpolation necessary for the GDP data? This seems to introduce more problems than it solves. I know that monthly indices of industrial production are available for Russia; these could be used as a proxy.

2) The model that is used does appear to be a little simple, omitting other key drivers of exchange rates. What about interest rates, investor expectations, or global risk?

3) I have never seen the abbreviation "ARDLBT" used, in hundreds of papers. The correct abbreviation is "ARDL." Also,what are the ARDL orders?

4) I have noticed evidence of a "cut and paste," and upon further research have found other papers with identical language.

5) The formatting of the paper needs to be improved.

Author Response

Dear Referee,

We authors thank you for your comments.

Please find our responses to your comments in red.

1) Why is interpolation necessary for the GDP data? This seems to introduce more problems than it solves. I know that monthly indices of industrial production are available for Russia; these could be used as a proxy.

Your suggestion of using industrial production was very useful and thank you for that. Following the suggestion, we have used producer/industrial production index and it makes more sense and provides better results.

2) The model that is used does appear to be a little simple, omitting other key drivers of exchange rates. What about interest rates, investor expectations, or global risk?

Thank you for your comment. In the theoretical framework section, we now discuss that the framework that we are using in the paper is borrowed from Chashin et al. (2004) and Habib and Kalamova (2007). In the seminal paper by Chashin et al (2004), a two-variable exchange rate equation for commodity-exporting countries has been conceptually adopted. Following the same conceptual framework, Habib and Kalamova (2007) have developed the framework, where the exchange rate is a function of oil price and productivity differential, for oil exporting economies.

Please kindly note that, other studies have also used such a parsimony specification for examining exchange rate in oil exporting developing economies: Hasanov (2010) for Azerbaijan, Korhonen and Juurikkala (2009) for 14 oil-exporters including Russia; Egert (2009) for the former Soviet Union countries including Russia, Kazakhstan and Azerbaijan, Kaplan and Aktash (2016) for five countries including Russia. Even, Benedictow et al. (2010), Ito (2010) for Russia, Jahan-Parvar and Mohammadi (2008) for 14 oil-exporters including Russia, Nikbakht (2009) for seven OPEC members, Ahmad and Hernandez (2013) for 12 oil producers and consumers have used only oil price to explain exchange rate in their empirical analyses.

Additionally, such a simple framework would be preferable when the number of observations is small, which is the case in our study.

Following your recommendation, we have added other explanatory variable in our analysis. We have included net foreign assets in our main specification and tested whether it can provide additional explanatory power in explaining behaviors of the real effective exchange rate in the countries under consideration, i.e., Russia, Kazakhstan, and Azerbaijan. Estimation results show that for the latter two countries the variable is insignificant with a negative sign, while it significantly distorts the coefficients of oil price and productivity differential in the Russian case. Therefore, as a research decision, we excluded it and consider the exchange rate as a function of oil price and relative productivity. Habib and Kalamova (2007) point out that terms of trade improve, net foreign assets increase, and government expenditure expands when the oil price rises and vice-versa. Moreover, Mironov and Petronevich (2015) show that productivity and net foreign assets are perfectly collinear for Russia. The same thing might be the case between net foreign assets and oil price for Russia and other countries considered here as well. Please kindly see the newly added subsection 5.1.Robustness check for the detailed discussion of the testing additional explanatory variable.

3) I have never seen the abbreviation "ARDLBT" used, in hundreds of papers. The correct abbreviation is "ARDL." Also, what are the ARDL orders?

Thank you for your remark. We have replaced the abbreviation "ARDLBT" to “ARDL”.  

Please kindly note that the ARDL orders are (3,0,0), (1,0,1) and (4,4,3) for Russia, Kazakhstan, and Azerbaijan, respectively.

4) I have noticed evidence of a "cut and paste," and upon further research have found other papers with identical language.

Thank you for raising this issue. The issue was related to the sub-section 3.1.Real Exchange Rate Equation, which we have changed completely in the revised version of the paper.

5) The formatting of the paper needs to be improved.

Thank you for commenting on the format of the paper. We tried to change formats of tables and figures as well as equations in the paper to make them more readable. In this regard, we have restructured the Literature Review section by presenting the reviewed papers in a table format. Additionally, we have excluded the former Table 2. Statistics for choosing an optimal lag size in order to have a better flow for a reader. Moreover, we have changed the former Table 4. The final ARDL Specifications by adding long-run coefficients, the speed of adjustments and diagnostics and other tests results, which would be more interesting for a reader. Finally, we have deleted equations (9)-(11) since we reported the long-run coefficients in the table mentioned above.

For robustness check, we added DOLS estimations results to Table 4. We have added the subsection 5.1. Robustness check to the section 5. Empirical Estimations and Discussion.

Please note that we have tried to improve the Introduction, Literature Review, Conclusion sections as well as have re-structured the general research design of the paper.

Finally, please kindly note that the manuscript has been extensively edited for the English language by a native speaker who has an expertise in the area investigated in our study.

Reviewer 2 Report

This paper analyses the impact of oil price shocks on real exchange rate for a sample of three CIS oil‐exporters (Russia, Kazakhstan and Azerbaijan) over the period period of 2004Q1‐2013Q4. To this end, the econometric methodology is based on the Autoregressive Distributed Lag Bounds Testing method. To estimate the real exchange rate, the author uses the behavioral equilibrium exchange rate (BEER) framework. Estimation results indicate that the oil price is one of the drivers of appreciation of real exchange rate of the selected economies in the long-run. In addition, the paper finds that productivity gains lead to the real appreciation. 

The paper is interesting but it investigates a topic largely explored in the literature. From this standpoint, the author must better stress the added value of this submission.

The section dedicated to the literature review must be reorganized in order to improve its analytical dimension. Specifically, in the current version of the submitted paper, the author presents different papers without a clear logic in their order. I suggest to arrange the papers, for example, by distinguish between econometric methods. It is important that the author stresses the main drawbacks of theses papers in order to highlight the main advantages of the econometric approach used in the paper. In addition, it lacks a presentation of the literature on currency commodities. This literature is very important in the context of the topic analyzed in this submitted paper.

If the choice of the behavioral equilibrium exchange rate (BEER) framework is relatively usual in the related literature, it seems to me that the authors could devote more attention about the main motives explaining such choice. In addition, the paper does not explain why the productivity variable is approximated by the GDP per capita.  Alberola et  al.  (1999)  use as a  proxy  the  ratio  of  the  consumer  price  index  (CPI)  to  the producer  price  index  (PPI). This point is particularly important as the GDP per capital had been interpolated to obtain quarterly data. The accuracy of this variable is debatable.

The model contains only two explanatory variables. I suggest that the author presents some tests to show that the exclusion of terms of trade and net foreign assets positions is relevant.

Figures 2 and 3 exhibit major changes in the time profiles of the variables used in the paper. These changes suggest the presence of structural breaks. As a consequence, instead of investigating the stationarity of the variables with ADF unit roots tests, it seems to more relevant to use the Zivot and Andrews (1992) approach. I think that introducing and exogenous dummy variable is not enough. 

In relation with the recent literature, it is important to distinguish between the sources of oil shocks (Kilian, 2009) to assess the impact of oil prices shocks on real exchange rate (see, for instance, Allegret et al. in Applied Economics 2017).

Finally, considering policy implications, it is important that the author considers the role of monetary regimes in the relationships between real exchange rate and oil prices. In addition, implications for fiscal policy devote more attention, notably from the point of view of fiscal rules.

Author Response

Dear Referee,

We authors thank you for your comments.

Please find our responses to your comments in red.

1) The paper is interesting but it investigates a topic largely explored in the literature. From this standpoint, the author must better stress the added value of this submission.

Thank you for raising this issue. In the revised version of the paper, we have tried to clearly express the main contributions of the study.

2) The section dedicated to the literature review must be reorganized in order to improve its analytical dimension. Specifically, in the current version of the submitted paper, the author presents different papers without a clear logic in their order. I suggest to arrange the papers, for example, by distinguish between econometric methods. It is important that the author stresses the main drawbacks of these papers in order to highlight the main advantages of the econometric approach used in the paper. In addition, it lacks a presentation of the literature on currency commodities. This literature is very important in the context of the topic analyzed in this submitted paper.

First of all, we thank you for your comment which was useful for improving the Literature Review section of our paper. By following your suggestion, we have restructured the Literature Review section by presenting all the reviewed papers in a table with chronological order. Now, one can clearly notice countries considered, time periods selected, methodologies employed and even numerical values obtained in the reviewed studies.

3) If the choice of the behavioral equilibrium exchange rate (BEER) framework is relatively usual in the related literature, it seems to me that the authors could devote more attention about the main motives explaining such choice. In addition, the paper does not explain why the productivity variable is approximated by the GDP per capita. Alberola et al.  (1999)  use as a  proxy  the  ratio  of  the  consumer  price  index  (CPI)  to  the producer  price  index  (PPI). This point is particularly important as the GDP per capital had been interpolated to obtain quarterly data. The accuracy of this variable is debatable.

We greatly value your suggestion. By following your suggestion and Alberola et al. (1999), we have used the ratio of the consumer price index to the producer price index as a proxy for productivity differential for all the three countries (please see 4.1.Data sub-section). Now, estimation results are more robust than they were before using the interpolated GDP per capita.

4) The model contains only two explanatory variables. I suggest that the author presents some tests to show that the exclusion of terms of trade and net foreign assets positions is relevant.

Thank you for your comment. Following your recommendation, we have added other explanatory variable in our analysis. Precisely speaking, we have included net foreign assets in our main specification and tested whether it can provide additional explanatory power in explaining the behaviors of the real effective exchange rate in the countries under consideration, i.e., Russia, Kazakhstan, and Azerbaijan. Estimation results show that for the latter two countries the variable is insignificant with a negative sign, while it significantly distorts the coefficients of oil price and productivity differential in the Russian case. Therefore, as a research decision, we excluded it and consider the exchange rate as a function of oil price and relative productivity. Habib and Kalamova (2007) point out that terms of trade improve, net foreign assets increases, and government expenditure expands when the oil price rises and vice-versa. Moreover, Mironov and Petronevich (2015) show that productivity and net foreign assets are perfectly collinear for Russia. The same thing might be the case between net foreign assets and oil price for Russia and other countries considered in the paper as well. Please kindly see the newly added sub-section, namely 5.1.Robustness check for the detailed discussion of the testing the additional explanatory variable.

In the theoretical framework section, we now discuss that the framework that we are using in the paper is borrowed from Chashin et al. (2004) and Habib and Kalamova (2007). In the seminal paper by Chashin et al (2004), a two-variable exchange rate equation for commodity-exporting countries has been conceptually adopted. Following the same conceptual framework, Habib and Kalamova (2007) have developed a framework, where the exchange rate is a function of oil price and productivity differential, for the oil exporting economies.

Please kindly note that, other studies have also used such a parsimony specification for examining exchange rate in oil exporting developing economies: Hasanov (2010) for Azerbaijan, Korhonen and Juurikkala (2009) for 14 oil-exporters including Russia; Egert (2009) for the former Soviet Union countries including Russia, Kazakhstan and Azerbaijan, Kaplan and Aktash (2016) for five countries including Russia have used two regressors in their studies where one of them is oil price. Even, Benedictow et al. (2010), Ito (2010) for Russia, Jahan-Parvar and Mohammadi (2008) for 14 oil-exporters including Russia, Nikbakht (2009) for seven OPEC members, Ahmad and Hernandez (2013) for 12 oil producers and consumers have used only oil price to explain the exchange rate in their empirical analysis.

Additionally, such a simple specification would be preferable when the number of observations is small, which is the case in our study.

5) Figures 2 and 3 exhibit major changes in the time profiles of the variables used in the paper. These changes suggest the presence of structural breaks. As a consequence, instead of investigating the stationarity of the variables with ADF unit roots tests, it seems to more relevant to use the Zivot and Andrews (1992) approach. I think that introducing and exogenous dummy variable is not enough. 

Your comment resulted in a deeper unit root analysis of the variables and we are grateful to you for that. We applied the Zivot and Andrews (1992) test to our variables and the test results show that they are non-stationary at the level form and stationary at their first differences. Zivot-Andrews test results have been added to the Table 2.

6) In relation with the recent literature, it is important to distinguish between the sources of oil shocks (Kilian, 2009) to assess the impact of oil prices shocks on real exchange rate (see, for instance, Allegret et al. in Applied Economics 2017).

Thank you for your comment. Please kindly note that the paper investigates the impact of oil price but not oil price shocks on the exchange rates of the selected countries. We, therefore, have not considered oil price shocks in our study but it can be an interesting topic for future research.

7) Finally, considering policy implications, it is important that the author considers the role of monetary regimes in the relationships between real exchange rate and oil prices. In addition, implications for fiscal policy devote more attention, notably from the point of view of fiscal rules.

Your comment was very helpful in re-shaping the sections 5.Empirical Estimations and Discussion and 6.Concluding Remarks by discussing fiscal policy measures that should be taken by the policy makers in the selected countries.

Please note that we have tried to improve the Introduction, Literature Review, Conclusion sections as well as have re-structured the general research design of the paper.

Finally, please kindly note that the manuscript has been extensively edited for the English language by a native speaker who has an expertise in the area investigated in our study.

Round 2

Reviewer 1 Report

The authors have taken efforts to address all the concerns.

Reviewer 2 Report

no additional comment